# Trauma-Focused Tuning in to Kids: Evaluation in a Clinical Service

**DOI:** 10.3390/children8111038

**Published:** 2021-11-11

**Authors:** Sophie S. Havighurst, Jessica L. Murphy, Christiane E. Kehoe

**Affiliations:** 1Mindful: Centre for Training and Research in Developmental Health, Department of Psychiatry, University of Melbourne, Melbourne, VIC 3010, Australia; ckehoe@unimelb.edu.au; 2Australian Childhood Foundation, Melbourne, VIC 3134, Australia; dr_jessicamurphy@ymail.com

**Keywords:** *Tuning in to Kids*, complex trauma, emotion coaching, emotion socialization, parent-child relationship, behavioral difficulties

## Abstract

This study evaluated the *Tuning in to Kids* (TIK) parenting program delivered in a clinical setting with 77 parents and caregivers (hereafter referred to as “parents”) of children who had experienced complex trauma. The TIK program targets parent emotion socialization to improve children’s emotional and behavioral functioning. The study utilized a single-group design with pre- and post-intervention measures. Seventy-seven parents of children (aged 3–15 years) who had experienced complex trauma completed a ten-week version of the Trauma-Focused *Tuning in to Kids* program (TF-TIK). Measures examined parent reports of: emotion socialization; parent-child relationship; parent mental health; children’s emotional and behavioral functioning. Parents reported significantly improved emotion socialization, parent-child relationship, parent mental health, as well as child emotion regulation and behavior. This study provides initial support for the use of the TF-TIK parenting program in a clinical setting with parents of children who have experienced complex trauma in order to prevent or reduce problems.

## 1. Introduction

The harmful psychosocial consequences for children who experience complex trauma are well documented, including adverse outcomes in social, behavioral, emotional, and physical development e.g., [1]. Complex trauma can be defined as exposure to multiple or prolonged traumatic events such as psychological maltreatment, neglect, physical and sexual abuse, and domestic violence that is chronic, begins early in childhood, and occurs within the primary caregiving system [2]. Of particular concern is the impact complex trauma has on the development of children’s emotional competence, including skills in identifying, understanding and regulating emotions e.g., [3]. Parents and carers (hereafter referred to as “parents”) of children exposed to complex trauma are more likely to use emotionally dismissive and/or harsh parenting, to have poor mental health and have fewer skills to assist children to learn about their emotions [4,5]. Poor parental mental health may contribute to limited emotional responsiveness, even after parents take part in parenting interventions [4,6]. Children exposed to complex trauma often do not develop emotional competence, which both limits their capacity to work through traumatic experiences and places them at greater risk for social, emotional, and behavioral difficulties [7]. Providing children are safe and with parents/carers that are able to be engaged in intervention, efforts to promote their emotional competence and assist them to work through the traumatic experiences are essential [8]. Programs that help parents with their own emotion regulation and teach skills in responding supportively to children’s emotions (components of what is often called emotion socialization [9]), have been found central for promoting children’s emotional competence [10], and may be effective components of intervention for children who have been exposed to complex trauma. This paper explores whether an emotion socialization parenting program, that has evidence with other community and clinical populations, is clinically effective with families where children have experienced complex trauma.

### 1.1. Background

There are a small number of evidence-based parenting interventions for children exposed to complex trauma [11], with limited research evaluating their benefits. Traditionally, parents are referred to generalist behavioral programs that teach parents to increase positive reinforcement and use consequences for misbehavior e.g., Triple P, [12]. While these programs have efficacy in reducing challenging behaviors in targeted and general populations [13], evidence where complex trauma is present is limited, especially in highly-stressed families, where there is maternal depression or marital violence [11]. Behavioral parenting interventions are less effective in reducing children’s internalizing difficulties e.g., anxiety or depression [14] and have lower efficacy for improving parents’ responses to children’s emotions when parents have poorer mental health [15].

Trauma-Focused Cognitive-Behavioral Therapy TF-CBT [16] is one intervention targeted at parents that has demonstrated positive results for children who have experienced trauma, including reductions in post-traumatic stress symptoms over 6 and 12-months follow up [17]. Work with parents/caregivers, however, is by helping the child to develop a narrative about their abuse experience using an exposure with response prevention model, rather than targeting parent emotion socialization as a primary focus so that children can safely develop skills in understanding and regulating their emotions within the context of a supportive, validating emotional environment. Extension work to TF-CBT includes the addition of emotion-focused parent work in the Let’s Connect program [18], however, evaluation with this addition for working with families exposed to trauma is still underway. 

Parenting interventions that more specifically target emotion socialization with families exposed to complex trauma have been reported. A number teach “emotion coaching”, a style of responding to children’s emotions that has been found optimal for children’s healthy emotional development [19]. Emotion coaching involves five main steps: (1) becoming aware of the child’s emotion, especially if it is at a lower intensity; (2) viewing the child’s emotion as an opportunity for intimacy and teaching; (3) communicating understanding and acceptance of emotions with empathy; (4) helping the child to use words to describe how they feel; before (5) assisting the child with problem solving or setting limits. Reminiscing and Emotion Training RET [20] teaches emotion coaching to parents where there has been maltreatment. A randomized controlled trial was conducted with 248 children between the ages of 3- to 6-years-old and their mothers, of which 165 were from maltreating families (randomized into RET or case management/provision of written parenting information) and found mothers who participated in RET were significantly better able to engage in elaboration about emotions and sensitive guidance of the child during reminiscing than mothers who received case management/written advice. Their children also demonstrated significantly better memory and emotion knowledge relative to controls. Katz and colleagues’ [21] twelve-week group parenting intervention for survivors of intimate partner violence addressed parent emotion awareness and regulation in early sessions before teaching emotion coaching skills in later sessions. In a pilot study with 50 mothers assigned to intervention or control, those receiving the intervention had improved parent emotion regulation on parent-report and physiological measures of respiratory sinus arrhythmia; improved observed and reported parenting and decreased parent-reported child depressive symptoms and decreased observed child negativity relative to controls who did not show these changes. The study provides preliminary evidence that teaching parents to better manage their own emotions as well as use of emotion coaching skills may be effective to reduce the negative impact of family violence and complex trauma. 

*Tuning in to Kids* (TIK) is another group parenting program that teaches parents skills in emotion awareness and regulation as well as skills in emotion coaching [22]. Based on emotion socialization theory [23], the program explores parents’ family of origin experiences with emotions in order to shift harsh and/or dismissive beliefs and reactions to emotions. Parents are instead taught skills that enable them to use the five steps of emotion coaching with their children when emotions occur. In particular, they learn: how to better identify emotions in their children, especially those that might underlie challenging behaviors; to view emotions as an opportunity to connect with children in order to help children understand and manage emotions; to develop skills in understanding children’s perspectives and responding with empathy to children’s emotional experience (rather than being judgmental); to either reflect or name emotions so children can put words to their feelings; to assist with solving problems, teach skills in emotion regulation, and, if necessary, look at limits around children’s behavior. Parents are encouraged to focus on noticing and empathizing with emotions, prior to problem solving and limit setting. This helps create an emotionally safe connection and builds children’s skills in emotional competence over time, assisting in reducing behavioral difficulties. The program also helps parents learn skills in regulating their own emotions, including engaging in self-care and learning how to manage strong emotions such as fear and anger. The program has produced promising results in clinical [24,25] and community samples [22,26] with positive impacts on parenting (e.g., reducing emotional dismissiveness and increasing emotion coaching) and children’s functioning (e.g., improvements in emotional competence and behavior). TIK has not yet been evaluated with parents of children exposed to complex trauma, however, given previous research highlighting the benefits of targeting emotion socialization to improve parental functioning, parent-child communication, and child emotional competence, the program may be suitable for this population. Exploring the benefits of an adapted version the evidence-based TIK program delivered in a real-world clinical setting is an important step in the scaling-out of an intervention such as this to novel populations [27]. TIK, rather than its variant for teenagers, was used because children were primarily younger and a focus on the core competencies, rather than additional adolescent-focused components (e.g., learning about adolescent development) was deemed appropriate. Children attending this service were often delayed in their emotional development due to their history of trauma and family difficulties. Clinicians who assessed each child and their family, referred the parents/carers to the program even if their children were in the early adolescent years because they were perceived to need basic emotion coaching skills that were part of the TIK program. 

### 1.2. Aim and Research Questions 

The current study aimed to evaluate an adapted version of the evidence-based *Tuning in to Kids* program with parents of children who had experienced complex trauma, including parents who had been involved in children’s abuse experiences. Working with these families can be very challenging and this population are not easy to engage in research, therefore, all parents in the study received the intervention. Based on previous research with TIK, it was hypothesized that the program would improve; (i) parent emotion-socialization (reducing emotion dismissing and increasing emotion coaching and empathy) and the parent-child relationship; (ii) parent mental health; (iii) children’s emotion regulation and behavior. 

## 2. Method

### 2.1. Participants

Participants were 77 parents (*M* = 35.1 years) of children (41 boys, 36 girls) aged 3 to 15 years (*M* = 9.57 years, *SD* = 3.08) recruited from a voluntary therapeutic service in Melbourne offering treatment for complex trauma, with all children displaying significant emotional and/or behavioral difficulties warranting psychological intervention. Reasons for referral to the service included, experiencing sexual abuse, physical abuse, emotional abuse, neglect, witnessing family violence, or other complex trauma (for example, interrupted attachments) with the majority experiencing multiple forms of trauma rather than single incident trauma. Some children were also referred because they were engaging in problem sexual behaviors. At intake all children were administered a standard screening tool used in the service, the Trauma Symptom Checklist for Young Children TSCYC [28]; with 87% falling in the clinical range for at least one of the subscales. This scale is not, however, very sensitive to complex and relational trauma. Clinicians reported the type of trauma children had experienced which corroborated the detail from the TSCYC. Table 1 displays participant descriptive information that clinicians had gathered and shows that many children experienced emotional abuse/neglect (90.9%), family violence (80.5%) as well as other traumas. Multiple traumas were most common, with 96% of parents reporting to clinicians that their children had experienced more than one form of trauma. Children’s experiences of trauma had primarily occurred within the family context (either immediate or extended family). The parents participating in the intervention study varied in whether they had been involved in the trauma or not. Many had their own complex difficulties, including, but not limited to, their own trauma history (69%), mental health difficulties (56%), child protection involvement (31%), family court involvement (21%), and/or other additional stressors (45%). 

### 2.2. Procedure

The TIK program was delivered in a clinical service for children who had experienced complex trauma and where the therapeutic goals were recovery from trauma and assisting parents or carers in how they responded to the child’s emotions and behaviors. The program was advertised throughout the clinic and participants (parents and carers) were able to self-select (by expressing interest to their treating therapist) or were invited by their treating therapist. Participants were invited when this was recommended as part of an intake assessment or when the overall treatment goals for the child were increasing social, emotional and behavioral competencies. Following screening and clinical assessment, parents were referred to the program by their child’s case-manager (13% of children were receiving concurrent individual therapy). Pre-program interviews were conducted and participants were excluded from attending if, (a) it was not recommended as their overall treatment plan, (b) the timing did not fit with other therapeutic interventions (i.e., if individual therapeutic work needed to take precedence for safety reasons at the time of the group being conducted), or (c) they were not deemed suitable for a group setting (for example, if their own personal trauma was so overwhelming that they were unable to learn in a group environment, and/or, if they possessed personality characteristics that were likely to disrupt the functioning of the group). Three (4.5%) parents were excluded because of intellectual disability, acquired brain injury, or severe anxiety. Inclusion and exclusion criteria were predominantly clinically based, and the decisions made by the individual therapist and case manager.

Nine parenting programs were conducted over four years with an average of 7 parents per group. Eighty-two percent of parents completed the program (attended at least 80% of sessions), however not all went on to complete follow-up measures. Both multiple and single carers attended (11 children had two carers attending), however, data were only collected from the primary carer who completed questionnaires at pre- (Time 1) and post-intervention (Time 2, 53%). Independent samples t-tests showed no significant differences on any measures between completers and non-completers. This study was approved by the University of Melbourne Human Ethics committee (# 1136183) and participants gave consent for their data to be used for the purposes of research.

### 2.3. Trauma-Focused Tuning in to Kids Intervention

Trauma-Focused *Tuning in to Kids* (TF-TIK) is an extended version of the original TIK program delivered in ten, two-hour, weekly sessions by two facilitators (psychologists or social workers trained by the TIK developers), using a structured manual [29] for eight of the ten sessions. Two sessions were added to assist parents’ understanding of children’s presentations post-trauma and included information about attachment disruption and the impact of trauma on child functioning. The subsequent 8 sessions were from the original 6-session TIK program but extended over 8 sessions, see [25], and aimed to facilitate changes in parents’ responses to children’s emotions to improve emotional connection. Parents were taught to “emotion coach” their children via a series of exercises, role plays and DVDs. Emphasis was placed on parents becoming aware of their own/their children’s emotions, including physiological symptoms, with a focus on understanding the function of children’s behavior and emotions. Parents learned skills in regulating their own emotions, especially managing their anger. The program has flexibility in how the content is delivered, however, fidelity checklists are used in each session, enabling the facilitator to address any missed content in later sessions. The second author, who was a facilitator in each of the groups (and therefore not blind to the study hypotheses), completed fidelity checklists at the end of each session and these showed 100% of compulsory program content was delivered for all groups.

### 2.4. Measures

The Parent Emotional Style Questionnaire (PESQ, 22) is a 21-item self-report measure of parent emotion socialization. Adapted from the Maternal Emotional Style Questionnaire MESQ [30], it explores parents’ responses to their children’s emotions (rated on a 5-point Likert scale). The adapted measure includes 7 additional items about responding to children’s fears and is made gender neutral so that mothers and fathers can respond. The measure has shown good reliability and validity in community and clinical samples [25,31]. The 10-item emotion dismissing subscale includes items such as, “Childhood is a happy-go-lucky time, not a time for feeling sad or angry”. The 11-items emotion coaching subscale includes items such as, “When my child is sad, it’s time to get close”. Five PESQ items were selected from the emotion coaching subscale to measure empathy (e.g., when my child is angry, I take some time to try to experience this feeling with him/her) because it is a specific component targeted in the TIK program and so separate measurement of this aspect of parenting was of interest. The measure showed good internal consistency in the current study with Cronbach’s alphas of 0.83, 0.82, and 0.90, respectively, for Time 1, and 0.87, 0.86, and 0.85 for Time 2. 

The Parenting Relationship Questionnaire PRQ [32], is a 45-item parent report measure that was used to assess aspects of the parent-child relationship. Five of the seven subscales were used (Attachment, Communication, Discipline Practices, Parenting Confidence, and Relational Frustration). Items are measured on a 4-point Likert scale from never through to always. Items include, “When my child is upset, I can calm him or her”, “I know what my child is feeling”, and “I remain calm when dealing with my child’s behaviour”. The measure has reported good reliability and validity (ibid), including when used with similar population to the current study [33]. In the current study, the scale showed good internal consistency with Cronbach’s alphas of 0.87, 0.85, 0.93, 0.86, and 0.94 for the respective subscales at Time 1, and 0.89, 0.90, 0.93, 0.88, and 0.93 at Time 2. 

Parent mental health was assessed using the short form of the Depression Anxiety and Stress Scale DASS [34], a 21-item measure with three subscales (Depression, Anxiety, and Stress) that is suitable for use in clinical and non-clinical settings (ibid). Participants rate the frequency and severity of experiencing negative emotions over the previous week on a 4-point Likert scale from 0 (did not apply to me at all) to 3 (applied to me most of the time). Example items include, “I found it hard to wind down” and “I felt like I had nothing to look forward to”. The measure has been found to have good reliability and validity [35]. Cronbach’s alpha were 0.95 for the Total scale at Time 1, and 0.96 at Time 2. 

Two measures were used to examine child emotional and behavioral functioning. The Emotion Regulation Checklist ERC [36] was used to measure children’s emotional regulation. The ERC has two dimensions: 15 items assess emotion lability/negativity, which describes the child’s negative affect, inflexibility, and mood lability (e.g., “Can recover quickly from upset or distress”); and eight items assess emotion regulation, which measures appropriate emotion expression and regulation (e.g., “Is prone to angry outbursts/tantrums easily”). Response options for all items range from never (1) to almost always (4). The ERC has good convergent and divergent validity with community samples and clinical populations [36,37]. Cronbach’s alphas for the ERC were 0.89 for Lability/Negativity at Time 1, 0.84 at Time 2, and 0.65 for Emotion Regulation (ER) at Time 1, 0.77 at Time 2.

The Eyberg Child Behavior Inventory ECBI [38] was used to assess children’s behavior problems. The Intensity score (36 items) examines the frequency of challenging behaviors on a 7-point Likert scale ranging from 1 (never) to 7 (always). The ECBI has been used extensively in research, including when used with clinical populations and there are many studies demonstrating its validity and reliability e.g., [39]. Cronbach’s alphas for the ECBI Total Behavior Intensity were 0.97 at Time 1 and 0.95 at Time 2.

### 2.5. Data Analysis

Data were examined for missing values, normality, and outliers. There were no violations of assumptions for paired t-test analyses apart from a deviation from normality for the DASS scale which was transformed in subsequent analyses. Pearson-mean imputation was used to replace missing scale items with mean values, providing that at least 95% of the data were available [40]. Bivariate correlations were used to assess the relationship between child age and outcome variables. Paired samples t-tests were conducted for all variables and intention to treat analyses, using Time 1 data carried forward were used for those with missing Time 2 data. When outcomes did not hold these are reported. 

## 3. Results

Descriptive statistics, *t*-tests, and effect sizes are reported in Table 2. There were no significant differences in outcomes for those also receiving individual therapy versus those who did not. There were no significant differences between those who completed data at Time 2 and those who did not on any of the demographic, clinical, or outcome variables. At Time 1, child age was correlated with parent empathy (*r* = −0.24, *p* = 0.046) indicating that as child age increased parents reported lower empathy. No other significant correlations were found between child age and any of the baseline or follow-up outcomes. Child age was also not correlated with change scores. 

### 3.1. Main Outcomes: Parenting and Parent Functioning

Parents reported a significant improvement in emotion coaching with their children (*d* = 0.44) and a significant decrease in emotion dismissing (*d* = 0.49) as measured by the Parental Emotion Style Questionnaire following the intervention. On this measure, parents also reported a significant improvement in their empathy towards their child (*d* = 0.77). Parents reported a significant improvement in their relationship with their child post intervention on the Parent Relationship Questionnaire. That is, they reported a significant improvement in their ability to understand and respond to their child’s emotional presentation, as measured by the Attachment subscale (*d* = 0.35), however in intention to treat analysis this outcome became non-significant (*p* = 0.09, *d* = 0.26). Parents also reported a significant improvement in their communication with their child (*d* = 0.37), as well as in their parenting confidence (*d* = 0.36). No significant differences were noted in parent reported discipline practices. Parents reported a significantly decreased level of relational frustration with their child (*d* = 0.52). Post-intervention, parents reported a significant reduction in their own mental health symptoms (DASS) total scores (*d* = 0.54).

### 3.2. Main Outcomes: Child Emotional and Behavioral Functioning

Following the intervention, parents reported their children to be significantly less labile in their emotions (*d* = 0.45) and better able to regulate their emotions (*d* = 0.15). They also reported that their children had significantly fewer behavioral difficulties (*d* = 0.39).

## 4. Discussion

This study evaluated the Trauma-Focused *Tuning in to Kids* (TF-TIK) program, a program targeting parents emotion socialization including their emotion coaching practices and their own emotion regulation, in a clinical setting with parents of children who had experienced complex trauma. The study included a highly troubled sample representing a “real world” group of children who had experienced complex trauma and their carers. Participants with these difficulties are often reluctant to engage in research and their problems are very difficult to treat. Results from TF-TIK were promising, with significant improvements in nearly all aspects of parent-reported parent and child functioning.

First, we examined the impact of TF-TIK on parent emotion socialization and the parent-child relationship. Consistent with our hypothesis, following the program parents reported increased emotion coaching and empathy, and less dismissiveness. The program provided parents with opportunities to view children’s emotions as a time for closeness and teaching (rather than avoiding or controlling emotions). This can be difficult for this population where traumatized individuals often rely on avoidant or dismissive strategies to manage emotions [41]. Parents also reported more connected parent-child relationships with significant improvements in their attachment, communication and confidence and less frustration with their child. It is possible that the emphasis on understanding the emotions behind children’s behavior allowed parents to develop empathy, leading to stronger parent-child relationships. Better skills in awareness and regulation have been found to be related to closer and more affectionate relationships [42].

The TF-TIK program does not teach behavior management strategies, favoring instead teaching parents to respond to emotions behind challenging behaviors, problem solving with their child, and setting limits. Post-intervention there were no significant changes in parenting Discipline yet child behavior significantly improved. These findings suggest behavioral techniques are not essential for changing child behavior, particularly when behavior may be trauma-related and helping the child process emotions may be important.

Our second hypothesis, which was confirmed, was that the program would lead to improvements in parent mental health. The TF-TIK program taught parents skills in emotional awareness, self-care, and anger management. In combination with improved parenting skills these may have contributed to less reactivity, greater emotional responsiveness, and more positive interactions with children. The program also provided parents with opportunities to examine their own childhood experiences with emotions in order to change parenting, a factor others have found necessary for breaking intergenerational patterns of dysfunctional parenting [43]. The combination of parents developing skills in responding to their children’s emotional needs along with parents exploring their own family histories with emotions in the group, appears to have contributed to parents experiencing less stress and greater confidence in parenting. Katz and colleagues [21] also found working with parent emotion awareness and regulation was a vital part of their intervention that occurred prior to mothers learning emotion coaching skills with their children. Further, we have previously found that when parents had more emotional difficulties themselves, greater improvement in child behavior occurred when parents participated in TIK than for those parents who received a behavioral parenting intervention [15]. We believe the parallel of targeting parent emotion awareness and regulation as well as parent emotion coaching may be especially important for those families where the parent is having significant difficulties themselves. 

Consistent with our final hypothesis, post-intervention, parents reported their children to have significantly better emotion regulation, and lower emotional lability/negativity and behavior problems. Although the program did not directly target behavior problems, focusing on the emotions that underlie children’s behavior (such as fears and worries) may have contributed to these improvements. If a child feels as though their emotions are being effectively responded to, they may be less likely to escalate their emotions and behavior. By reducing children’s negativity/lability and improving the relationship with their caregiver, children may be better able to build a repertoire of competencies influencing their behavior [42]. Parent responsiveness to children’s emotions may also have translated to an improved emotional climate at home, which is likely to positively impact children’s behavior [44]. The effect sizes in change on the child variables were small. This is consistent with other research with TIK. Given the complexity in the presentation of these children, further clinical treatment was most often needed. Many required ongoing individual therapy to work through their traumatic experiences, and their families needed support (often long-term) to help them ensure the care of their children was safe, secure and fostered recovery. This is consistent with findings from other research trials of treatments for children exposed to complex trauma [45]. Future research using TIK in a multi-systemic intervention to address these complexities, would be useful. 

### Limitations and Future Research

As a study with a clinical population there were limitations: the absence of a randomized control group; significant attrition with follow-up questionnaires (although 82% of parents completed the program); no data on those who refused participation; dependence on parental report; with only a short follow-up. Finally, effect sizes for child outcomes were relatively small post-intervention and longer follow-up may have revealed greater improvement. Future research with a larger sample, a control group and using observation methods of evaluation is warranted, however, it is acknowledged this can be very difficult with families where there is complex trauma that may still be occurring. A larger sample would, however, enable testing of the mechanisms of change occurring or whether there were subgroups of families who benefitted more (or less) from the intervention.

## 5. Conclusions

This study with a clinical population provides preliminary support for TF-TIK with parents and carers of children who have experienced complex trauma. To date, few parenting programs have proven successful with this population. The current study demonstrated that a modified evidence-based parenting program was able to be effectively implemented in a clinical setting and appears to have led to positive changes in parent-child relationships, parent empathy, mental health and confidence, and lower levels of child behavior problems. An evidence-based parent emotion-socialization program is a valuable addition to the available treatments for complex trauma and may complement individual treatments such as Trauma-Focused CBT [16]. 

## Figures and Tables

**Table 1 children-08-01038-t001:** Participant descriptives (N = 77).

Characteristic	*n*	%
Child age group		
3–6 years	11	14.3
7–10 years	32	41.6
11–15 years	34	44.2
Caregiver gender (female)	68	88.3
Type of caregiver ^a^		
Biological parent	58	75.3
Carer ^b^	9	11.7
Grandparent	4	5.2
Foster parent	4	5.2
Income of caregiver family ^a^		
Low (under $35,000)	33	42.9
Medium ($35,000–$75,000)	16	20.8
High (75,000+)	14	18.2
Role in child’s trauma ^a,c^		
Engaged in abuse/neglect of their child	16	20.8
Witnessed abuse/neglect and did not intervene	16	20.8
Witnessed abuse/neglect and intervened	7	9.1
Had no role in the abuse/neglect of the child	14	18.2
Type of trauma experienced by child/indication of problems ^a,d^		
Multiple trauma	67	87.0
Psychological/Emotional abuse/neglect	70	90.9
Family violence	62	80.5
Environmental neglect	24	31.2
Physical abuse	27	35.1
Sexual abuse	9	11.7
Problem sexual behavior/sexually abusive behavior ^e^	35	45.5
Stressors affecting participant during parenting program ^a^		
Caregiver mental health difficulties ^f^	27	35.1
Intergenerational trauma ^g^	53	68.8
Child protection service involvement ^c^	24	31.2
Family court ^c^	16	20.8
Additional stressors (e.g., abuse of child, custody issue) ^c^	35	45.5

*Notes.* ^a^ Missing data for this variable; ^b^ Carer includes kinship carers and those who have permanent care of their non-biological child; ^c^ this variable was conducted through file review, however, for 24 participants this information was not available. ^d^ As assessed by child’s therapist in clinical interview who assessed for all types of abuse-related trauma; ^e^ Children met this criterion if they were referred for trauma and also engaged in problem sexual behavior or sexually abusive behavior with others; ^f^ Participants met this criterion if they were rated as “moderate” or above on at least one scale of the Depression Anxiety and Stress Scale; ^g^ Participants who had disclosed their own trauma history over the course of their involvement with the service were categorized as having experienced intergenerational trauma.

**Table 2 children-08-01038-t002:** Intervention outcomes.

	Time 1			Time 2					
Variable	*M*	*SD*	*n*	*M*	*SD*	*n*	*t*	*p*	*d* ^a^
Parent Emotion Socialization									
Emotion Coaching	41.63	7.28	72	44.60	6.01	40	−3.00	0.005	0.44
Emotion Dismissing	36.74	6.03	69	33.63	6.54	40	3.07	0.004	0.49
Empathy	17.54	3.73	72	20.43	3.37	40	−4.11	0.000	0.77
Parent-Child Relationship									
Attachment	40.68	11.16	72	44.73	12.26	40	−3.37	0.002	0.35
Communication	40.22	12.16	64	44.64	11.88	36	−3.12	0.004	0.37
Discipline Practices	43.54	11.58	72	40.13	10.10	40	−1.23	0.226	0.31
Parenting Confidence	38.44	12.48	72	42.95	12.59	40	−3.47	0.001	0.36
Relational Frustration	61.89	15.22	72	54.78	11.91	40	3.50	0.001	0.52
Parent Mental Health Total ^b^	36.35	26.91	68	26.05	26.58	40	3.71	0.001	0.54
Child Emotional Competence									
Lability/Negativity	32.96	6.07	71	30.12	6.47	41	2.50	0.017	0.45
Emotional Regulation	20.17	3.27	71	20.66	3.07	41	−3.16	0.003	0.15
Children’s Anxiety and Behavior									
Total Anxiety	26.08	17.47	64	24.61	15.45	36	1.63	0.113	0.09
Total Behavior Intensity	129.73	46.75	73	113.41	36.27	41	2.25	0.030	0.39

*Note.* ^a^ Cohen’s d obtained using means and standard deviations for effect size: 0.2 = small; 0.5 = medium; 0.8 large; ^b^ Values shown are raw scores prior to transformation.

## Data Availability

Consent was not sought from participants for their data to be available.

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
