# Peer review of "Trauma-Focused Tuning in to Kids: Evaluation in a Clinical Service"

_children, 2021, doi:10.3390/children8111038_

Round 1

Reviewer 1 Report

I reviewed with interest this article TRAUMA-FOCUSED TUNING IN TO KIDS: EVALUATION 2 IN A CLINICAL SERVICE.

Considering the harmful consequences for children for experienced complex trauma is very important. All the paper is pleasant to read.

The introduction brings the question very clearly, according to the literature. In Methods, the study is explained with enough details to be reproduced. 

The data are able to test the original proposed hypotheses even there is only a single-group design.

One of the authors is involved in the tested program and it is a limit that is underlined by authors.

Authors conclusions are justified given the data: results are promising with positive impact of the TF-TIK for both parents/carers and children and this study is presented as a preliminary support for this new program.

Author Response

No revisions requested. 

Reviewer 2 Report

This interesting paper reports on pre-post assessment of implementation of an intervention for parents of traumatized children. It was implemented in real world clinical setting, and the authors are to be congratulated for managing this difficult task. The authors and their paper make the case for why an intervention such as TF-TIK should be implemented in settings that treat traumatized children and parents. And although the study uses exclusively parent report outcomes and uses participants as their own controls, this is acceptable for an initial demonstration of an adapted intervention’s effectiveness. Thus, the study is well justified.

There are, nonetheless, some concerns about the paper in its current form.

  1. The sample is described as children who have been exposed to “complex trauma.” This very term is controversial, and the data provided make it challenging for the reader to determine exactly who experienced what. Table 1 lists different types of exposure, including that 87% of the sample experienced “multiple traumas.” Does this means multiple types of abuse? Or would neglect (which is not trauma) be included? Is placement in foster care considered a trauma? In other words, for the reader to understand who received the intervention, a much more granular description of the sample is needed—this is essential for determining to whom results might be generalized.  If chronicity of abuse is included in the definitions, how is that defined?  And since the authors definition includes that the trauma occur within the primary caregiving system, how is the reader to understand that 18% had no role in the abuse/neglect of their child.  More details and more clarity is needed here.

  1. There are also many problems with the measures. Details about on whom the measures have been validated are mostly lacking. Also, there was little rationale as to why only some subscales were used but not others. Also, what is the justification for selecting 5 items from a scale to assess empathy? Merely demonstrating that the new scale has a decent alpha leaves many questions about what it measures unanswered.

  1. Some additional details about what the investigators mean by “emotion coaching” would be helpful to those readers who are less familiar with TF-TIK. What did parents do exactly to coach their children about emotions?

  1. There is a lack of developmental perspective that is apparent considering that participants range from 3 to 15 years. When their traumatic experiences occurred, how developmental differences may have affected the clinical picture, and how the intervention was applied to parents across this broad range of developmental capabilities is not really considered.

  1. I am unclear if the outcomes reported on p. 8 were a priori hypotheses? The end of the introduction mentions children’s emotion competence and behavior but on p. 8 the “main” outcomes reported are emotional lability, emotion regulation and behavioral difficulties. At a minimum, it would be helpful if constructs were labeled consistently for the reader. If the latter are different from the former, then perhaps the hypotheses being tested should be more clearly stated in the Introduction.

Author Response

Reviewers Comment: The sample is described as children who have been exposed to “complex trauma.” This very term is controversial, and the data provided make it challenging for the reader to determine exactly who experienced what. Table 1 lists different types of exposure, including that 87% of the sample experienced “multiple traumas.” Does this means multiple types of abuse? Or would neglect (which is not trauma) be included? Is placement in foster care considered a trauma? In other words, for the reader to understand who received the intervention, a much more granular description of the sample is needed—this is essential for determining to whom results might be generalized.  If chronicity of abuse is included in the definitions, how is that defined?  And since the authors definition includes that the trauma occur within the primary caregiving system, how is the reader to understand that 18% had no role in the abuse/neglect of their child.  More details and more clarity is needed here.

Our response: We agree that the term complex trauma includes a range of possible circumstances and we have tried to provide more detail to clarify and describe our participant sample.

On page 3 of the manuscript we firstly defined complex trauma: “Complex trauma can be defined as exposure to multiple or prolonged traumatic events such as psychological maltreatment, neglect, physical and sexual abuse, and domestic violence that is chronic, begins early in childhood, and occurs within the primary caregiving system [2].”

In our study the sample were from a service for children who had experienced complex trauma rather than single incident traumas. We have mentioned this in a number of places, and illustrate this by including detail in the table about the percentage of children/families who had reported to clinicians the types of trauma children had experience.

In addition, under participants detail section in the text (page 8) we have also added that this was not a service for single incident trauma.

We also added more detail to demonstrate the overlap in categories of trauma for children and explained more about the role of the parent in the trauma. “Clinicians reported the type of trauma children had experienced which corroborated the detail from the TSCYC. Table 1 displays participant descriptive information that clinicians had gathered and shows that many children experienced emotional abuse/neglect (90.9%), family violence (80.5%) as well as other traumas. Multiple traumas were most common, with 96% of parents reporting to clinicians that their children had experienced more than one form of trauma. Children’s experiences of trauma had primarily occurred within the family context (either immediate or extended family). The parents participating in the intervention varied in whether they had been involved in the trauma or not.”

Reviewers Comment: There are also many problems with the measures. Details about on whom the measures have been validated are mostly lacking. Also, there was little rationale as to why only some subscales were used but not others. Also, what is the justification for selecting 5 items from a scale to assess empathy? Merely demonstrating that the new scale has a decent alpha leaves many questions about what it measures unanswered.

Our response: We have added additional detail throughout the measures section on reliability and validity and the populations they have been used with (clinical populations). We have also added justification as to why specific subscales were chosen for use in particular the focus on measuring parent empathy.

Reviewers Comment: Some additional details about what the investigators mean by “emotion coaching” would be helpful to those readers who are less familiar with TF-TIK. What did parents do exactly to coach their children about emotions?

Our response: On page 5, a definition of emotion coaching is included. We have also added additional detail under the description of the TIK program on page 6.

“Parents are instead taught skills that enable them to use the five steps of emotion coaching with their children when emotions occur. In particular, they learn: how to better identify emotions in their children, especially those that might underlie challenging behaviors; to view emotions as an opportunity to connect with children in order to help children understand and manage emotions; to develop skills in understanding children’s perspectives and responding with empathy to children’s emotional experience (rather than being judgmental); to either reflect or name emotions so children can put words to their feelings; and to assist with solving problems, teach skills in emotion regulation, and if necessary look at limits around children’s behavior. Parents are encouraged to focus on noticing and empathizing with emotions, prior to problem solving and limit setting. This helps create an emotionally safe connection and builds children’s skills in emotional competence over time, assisting in reducing behavioral difficulties.”

Reviewers Comment: There is a lack of developmental perspective that is apparent considering that participants range from 3 to 15 years. When their traumatic experiences occurred, how developmental differences may have affected the clinical picture, and how the intervention was applied to parents across this broad range of developmental capabilities is not really considered.

Our response: Because of the complex nature of families presenting to this service, and because many of the adolescents were socially and emotionally delayed due to their experiences, the TIK program which is used for 3-12 year old children was deemed appropriate by the service. Both TIK and the Tuning in to Teens version of the program have core components of emotion coaching that are similar. We have added a statement in the text to address this:

“TIK, rather than its variant for teenagers, was used because children were primarily younger and a focus on the core competencies, rather than additional adolescent-focused components (e.g., learning about adolescent development) was deemed appropriate. Children attending this service were often delayed in their emotional development due to their history of trauma and family difficulties. Clinicians who assessed each child and their family, referred the parents/carers to the program even if their children were in the early adolescent years because they were perceived to need basic emotion coaching skills that were part of the TIK program.”

Reviewers Comment: I am unclear if the outcomes reported on p. 8 were a priori hypotheses? The end of the introduction mentions children’s emotion competence and behavior but on p. 8 the “main” outcomes reported are emotional lability, emotion regulation and behavioral difficulties. At a minimum, it would be helpful if constructs were labeled consistently for the reader. If the latter are different from the former, then perhaps the hypotheses being tested should be more clearly stated in the Introduction.

Our response: We have changed emotion competence to emotion regulation in the aims and the measures section and then specified under the measures section that emotion regulation included two components – lability and emotion regulation. This should clarify that these were a priori hypotheses.

Reviewer 3 Report

The manuscript is well writte. However, if possible, in Participants' Section ("referred because they were engaging in problem sexual behaviors") I suggest to specify what the authors mean with problem sexual behaviours. In addition, I suggest to the authors to indicate if the "diagnosis" of complex trauma of these patients was clinical or already judicially defined.

referred because they were engaging in problem sexual behaviors

Author Response

Reviewers Comment: The manuscript is well writte. However, if possible, in Participants' Section ("referred because they were engaging in problem sexual behaviors") I suggest to specify what the authors mean with problem sexual behaviours. In addition, I suggest to the authors to indicate if the "diagnosis" of complex trauma of these patients was clinical or already judicially defined.

Our response: We have added detail in Table 1 – Participant Descriptives to explain that these were children referred for trauma who also engaged in sexually abusive behaviours with others. Children met this criterion if they were referred for trauma and also engaged in problem sexual behaviour or sexually abusive behaviour with others”